# Quantitative modeling of signaling in aggressive B cell lymphoma unveils conserved core network

**Bertram Klinger**[1,2,3☯]*, **Isabel Rausch**[4,5☯], **Anja Sieber**[1], **Helmut Kutz**[6],
**Vanessa Kruse**[4], **Marieluise Kirchner**[7], **Philipp Mertins**[7], **Arnd Kieser**[6,8,9],
**Nils Blüthgen**[1,2,3☯]*, **Dieter Kube**[4☯]*

**1** Institute of Pathology, Charité-Universitätsmedizin Berlin, corporate member of Freie Universität Berlin and Humboldt-Universität zu Berlin, Berlin, Germany, **2** Institute for Theoretical Biology, Humboldt-Universität zu Berlin, Berlin, Germany, **3** German Cancer Consortium (DKTK) Partner Site Berlin, German Cancer Research Center (DKFZ), Heidelberg, Germany, **4** Clinic of Hematology and Medical Oncology, University Medical Centre Goettingen, Göttingen, Germany, **5** ZytoVision GmbH, Bremerhaven, Germany, **6** Research Unit Gene Vectors, Helmholtz Center Munich—German Research Center for Environmental Health, Munich, Germany, **7** Core Unit Proteomics, Berlin Institute of Health at Charité—Universitaetsmedizin Berlin and Max-Delbrueck-Center for Molecular Medicine, Berlin, Germany, **8** Research Unit Signaling and Translation, Helmholtz Center Munich—German Research Center for Environmental Health, Neuherberg, Germany, **9** German Center for Infection Research (DZIF), Partner Site Munich, Germany

☯ These authors contributed equally to this work.
* bertram.klinger@charite.de (BK); nils.bluethgen@charite.de (NB); Dieter.Kube@med.uni-goettingen.de (DK)

**Data Availability Statement:** Mass spec data are available via ProteomeXchange with identifier PXD047709. All relevant data and pipelines for modeling can be accessed on the zenodo

## Abstract

B cell receptor (BCR) signaling is required for the survival and maturation of B cells and is deregulated in B cell lymphomas. While proximal BCR signaling is well studied, little is known about the crosstalk of downstream effector pathways, and a comprehensive quantitative network analysis of BCR signaling is missing. Here, we semi-quantitatively modelled BCR signaling in Burkitt lymphoma (BL) cells using systematically perturbed phosphorylation data of BL-2 and BL-41 cells. The models unveiled feedback and crosstalk structures in the BCR signaling network, including a negative crosstalk from p38 to MEK/ERK. The relevance of the crosstalk was verified for BCR and CD40 signaling in different BL cells and confirmed by global phosphoproteomics on ERK itself and known ERK target sites. Compared to the starting network, the trained network for BL-2 cells was better transferable to BL-41 cells. Moreover, the BL-2 network was also suited to model BCR signaling in Diffuse large B cell lymphoma cells lines with aberrant BCR signaling (HBL-1, OCI-LY3), indicating that BCR aberration does not cause a major downstream rewiring.

## Author summary

B cell receptors bind specific antigens, and upon binding, they activate a signal transduction network which ultimately primes the cells for proliferation and affinity maturation. B-cell receptor signaling is often altered in B-cell lymphoma, leading to altered or chronic activation of the network. In this study we compared the signal transduction network

repository https://zenodo.org/doi/10.5281/zenodo.10732059.

**Funding:** This work was funded by grants from the Federal Ministry of Education and Research (BMBF) within the joint project e:Med MMML-Demonstrator (031A428B to DK, 031A428F to NB and 031A428G to AK), and MSTARS-2 (16LW0239K, 16LW0240) to NB and PM. BK was funded by Deutsche Krebshilfe (70114307), DKFZ (Young Investigator Grant) and Bundesinstitut für Risikobewertung (60-0102-01.P61). The funders had no role in study design, data collection and analysis, decision to publish, or preparation of the manuscript.

**Competing interests:** The authors have declared that no competing interests exist.

downstream of acute and aberrant B cell receptor activity in cell lines originating from Burkitt and diffuse large B-cell lymphoma, respectively. By applying kinase inhibitors, we measured phosphorylation state changes in the network nodes. Mathematical modeling revealed a 16-node core network conserved in cells with acute and abberant B cell receptor signaling. In the network we detected hitherto undescribed crosstalks and feedbacks, structures known to confer treatment robustness. We elucidated and verified a negative crosstalk between the mitogen-activated protein kinases (MAPK) p38 and ERK. We further discovered that the negative feedback from ERK to its upstream kinase RAF, which in solid tumors neutralizes treatments targeting ERK or MEK, is also present in B cells. Altogether these findings may inform future treatment strategies targeting overactive B cell receptor or help to explain treatment resistance.

## Introduction

Intracellular signaling pathways are central to the communication of a cell with its environment, and control many important cellular processes and fates. These pathways are often activated by ligands that bind to cognate cell surface receptors and activate a specific set of intracellular proteins, often by tyrosine or serine/threonine phosphorylation. In many disease contexts, these pathways are deregulated, for instance by mutations in key signaling proteins. Consequently, activation can occur independent of ligands or stimuli. Signaling pathways are embedded into complex networks with feedback and crosstalk. It is therefore difficult to predict how these networks change when a pathway is chronically activated, and how the pathway reacts upon targeted interference. Mathematical modeling of intracellular pathways based on systematic perturbation data is a valuable approach to disentangle the difference in signaling networks of ligand/stimuli-induced pathways vs. chronic (aberrant) active pathways. A better understanding of the interactions of chronically activated signaling pathways is important to improve the prediction and further design of targeted therapies.

B cell receptor signaling represents a network for which the extend of feedback and crosstalk still remains unclear. Furthermore, a better dissection of chronic vs. acute signaling will support our understanding of related diseases. In normal B cells, the pathway is triggered by antigens, but is chronically activated in specific sets of Non-Hodgkin B cell lymphoma (NHL). In normal physiology the binding of a specific antigen to the B cell receptor complex leads to a very fast rebuilding and recruitment of a number of proximal signaling molecules including the ITAM motif molecules CD79A/B, the kinases LYN/SYK and SLP65/Btk/PI3K [1–3]. The activated BCR subsequently recruits multiple downstream signaling molecules and pathways, some of which include those dependent upon phospholipase Cγ, Protein kinase C or RAF-MAPK, PI3K, GSK3, MTOR and NF-kB. In addition, also JNK and p38MAPK are activated downstream of the BCR [4]. These pathways converge in the activation of a set of different transcription factors controlling B cell proliferation and survival, including c-Myc, NF-AT, Elk1, c-Jun and ATF2 (4–6).

In human B cells the activation of the key signaling pathway RAF/MEK/ERK by BCR seems to be similar to the activation by receptor tyrosine kinases (Satpathy et al., 2015; Vanshylla et al., 2018) [5,6]. This signal transduction network downstream of the B cell receptor (BCR) has received much attention, as it is a major regulator in adaptive immunity [7–10]. It has been shown that the relative activity of the multiple pathways downstream of BCR determines the outcome of BCR signaling [11]. The activity of the BCR is important for the expansion and survival and can be supported by both antigen-dependent and antigen-independent

mechanisms [12]. Compelling evidence is provided that in different NHL subtypes aberrant BCR signaling activation is important [9,13–19]. BCR signaling is often studied in Burkitt lymphoma (BL) cell lines [5,6,12,20,21].

While the EGFR-signaling network or the T cell receptor pathway have been quantitatively studied in many contexts, the BCR signaling and its oncogenic variations have so far only gained limited attention. Elaborated network models of EGFR and T cell signaling include proximal and distal elements but also feedback loops [2,22–28]. Some of these models discovered and quantified pathway interactions and feedback loops, which have important implications also for clinical applications. These include positive feedback loops that strengthen the therapeutic success and negative feedback loops that lead to adverse events or even therapeutic failure [26,29–32]. For example, the somewhat disappointing response rates to drugs targeting the intracellular RAS/MAPK pathway could be explained by strong negative feedbacks [26,33]. Now it is becoming more and more clear that also in BCR signaling the signal is propagated through a complex context-dependent network. The complexity of this signaling network is also characterized by an interplay of RAF/MEK/ERK, JNK, p38, NF-kB and PI3K, originally described as parallel pathways of activation. However, there is a substantial crosstalk between these different downstream signaling molecules not yet characterized in sufficient detail. Reaching a better level of understanding will allow to answer questions such as whether these crosstalks are cell type specific and thus account in part for the differential activity of BCR and its functional outcome, but can also unveil therapeutic opportunities.

The few existing BCR signaling models are mostly deterministic models that include BCR signal propagation by a specific set of network nodes [23,24,34,35]. The models cover both membrane proximal, early signaling events and to some extend downstream signaling events. Furthermore, a detailed model of the feedback loops involving LYN and SYK incorporating every phosphorylation event for six proximal signaling components has been established [34]. However, an exhaustive quantitative modeling of downstream pathways is currently missing.

A previous study [36] attempted to infer an oncogenic signaling network in aggressive lymphoma cell lines indirectly from gene expression data following signaling perturbations using the Boolean Nested Effects Modeling (B-NEM) approach. By this simple modeling approach, an acyclic network with only activating interactions was derived which contained no feedbacks and lacked quantitative information. While such models help to gain the understanding of pathway interactions, they are not suitable for simulations or quantitative model comparisons.

To get a more fine-grained semi-quantitative understanding of the signaling network, we therefore decided to employ a modeling approach on the more information rich readout of phosphorylation data after systematic pathway perturbations which contains more predictive power [37]. We have previously developed an approach termed STeady-STate Analysis of Signaling Networks (STASNet) that is based on Modular Response Analysis (MRA) [38] and applied this tool to decipher EGFR/RAS signaling in different tumors [26,39–42] as well as to compare mouse embryonic stem cells with different sex chromosome compositions [43]. The main concept of MRA is that the measurable global response matrix **R** (e.g., log fold changes of steady state measurements before and after systematic single node perturbations for every node of the network) theoretically contains the information to derive the so-called local response **r**, a matrix whose non-zero entries quantify the edges of the underlying network structure which are called local response coefficients (see Material and Methods). A positive and negative response coefficient encodes for an increase or decrease of the downstream node's activity, respectively, where $|r| > 1$ amplifies, $|r| < 1$ dampens and $|r| = 1$ neutrally relays the upstream signal. In STASNet we adapted the MRA theory to real world applications to account for (1) measurement noise using a maximum likelihood and model selection strategy and (2) incomplete perturbation and measurement designs by adding a non-identifiability

analysis before parameter fitting [39]. Due to the latter, we summarize non-identifiable response coefficients into identifiable combinations thereof, and term those 'coefficient paths'. The advantage of the MRA approach is that it allows to model and identify feedback loops that are known to play a crucial role in conferring robustness, shaping the signaling dynamics or integrating multiple converging signals and thus are vital to consider when searching for drug combinations eligible for targeted therapeutics. Furthermore, in contrast to the previous Boolean approach, STASNet allows to semi-quantitatively (on steady state change level) compare signaling networks and to predict simulations across cell line models.

Based on systematic perturbation data we developed an MRA-based network model for acute signaling in a Burkitt lymphoma (BL) cell line BL-2. We show that this model is transferable to perturbation data of another BL cell line BL-41. This supports the view that our model reflects common elements of intracellular signaling of acute B-cell signaling. Additional phosphoproteomic analyses of selected interventions support key insights of this model. After establishing these quantitative network models, we found that the BL-2-derived network structure forms a solid base to describe signaling networks in cell lines from Diffuse large B cell lymphoma (DLBCL) with aberrant BCR signaling. Thus, by integrated phosphoproteomic analysis, we have uncovered a conserved core B cell receptor-regulated signaling network.

## Results and Discussion

To dissect B cell signaling in aggressive NHL cells and get better insights into corresponding oncogenic signaling networks, we followed an approach that we previously established for EGFR signal transduction networks in solid cancer cell lines [26,39]. This approach combines experimental quantitative perturbation data sets with a mechanistic computational modeling approach derived from MRA.

### Generation of signaling perturbation data set

We decided to generate the first model on the Burkitt lymphoma cell line BL-2 as this type of lymphoma is a role model for studying acute B cell signaling [21, 44]. In BL-2 cells an external stimulus ($\alpha$-IgM) is able to activate the BCR and its downstream intracellular signaling [44, 45]. Stimulation-mediated intracellular signaling typically displays a strong transient response with different kinetics followed by short or longer lasting activation plateau interval. Therefore, time-series experiments were performed to determine the optimal time point for the pathway intervention experiments. A strong response for AKT, ERK1/2, MEK, p70S6K and p38/MAPK14 over time in $\alpha$-IgM treated BL-2 cells is observed using both immunoblotting and a bead-based ELISA platform (**S1 Fig**). The 30-min time point was chosen for further experiments, as the interpretation of the modeling procedure requires the signaling network to be approximately in steady state.

To establish an information rich dataset for modeling, we combined stimulation of the receptor with inhibitions of several key signaling nodes with targeted small-molecule inhibitors. Specifically, we preincubated inhibitors for JNK (5µM SP600125), MEK (1µM AZD6244), PI3K (2µM BKM120), Btk (10µM Ibrutinib), AKT (1µM MK-2206), MTOR (1µM Rapamycin), IKK (10µM MLN120B) and p38 (2µM SB203580) or solvent control for 3h. We then stimulated BCL-2 cells with $\alpha$-IgM for 30min or left unstimulated (control). Before and after perturbation, phosphorylation of key signaling molecules was quantified to create a high-dimensional data set. A bead-based multiplex ELISA platform (MagPix) was then used to measure the phosphorylation of fourteen key signaling and effector proteins: SYK$^{Y352}$, ZAP70$^{Y319}$, Btk$^{Y223}$, AKT$^{S473}$, RPS6$^{S235/236}$, BAD$^{S136}$, ERK2$^{T185/Y187}$, MEK1$^{S217/S221}$, p90RSK$^{S380}$, GSK-3a/b$^{S21/S9}$, NF-κB-p65$^{S536}$, HSP27$^{S78}$, JNK$^{T183/Y185}$ and cJun$^{S63}$. The resulting data sets are

presented as a heat map of log2 fold changes compared to unperturbed and unstimulated controls, were we noticed a stark contrast in complexity between stimulated and unstimulated inhibitor response data (**Fig 1A**).

## Model-based analysis of B cell receptor signaling in BL-2 cells

We used the perturbation data set together with a literature-based starting network [1,12,40,46] for our modeling pipeline to reverse engineer the signaling network downstream of the BCR (**Fig 1B**). We performed modeling using STASNet, a Modular Response Analysis-based modeling R package, to estimate optimal coefficient composition and quantification for a given network structure to best match the perturbation data [Parametrization]. After fitting the initial network, we adapted the network structure to the specific cell system by removing links that did not contribute significantly to the goodness of fit (likelihood ratio test, p>0.05) and tested systematically if adding a link significantly improves the network fit (likelihood ratio test, p<0.05) [Adapt Network]. To prevent overfitting, structurally altered network models were validated by comparing simulations to data that were not used for model parameterization [Consistency]. These three modeling steps were iteratively repeated for every network change until no further alteration was supported by the data [Final Network]. **Fig 1C** shows the goodness of fit and the statistics of the consistency check during the network modeling steps. During the course of modeling the reduced chi-square statistics, representing goodness of fit, steadily decreases indicating better or equally good fit with every network alteration. Next to the model fit a simultaneous consistency check procedure is conducted assessing the ability to model the unseen less complex unstimulated inhibitor data as percent reduction of weighted sum squared residuals (WSSR), when compared to untreated data as null model. For each modeling step the model predicts unseen data better or equally well than for the previous step. This is a good indication that the model development did not lead to overfitting.

**Fig 1D** presents the literature-derived starting network adjusted to the structure of the final BCR signaling network model for BL-2 cells with pruned links indicated in grey and novel links indicated in green (see **S2 Fig** for model fit vs. data comparison). Four of the removed links correspond to redundancies in the receptor proximal signaling structure around PI3K where removal resolved this non-identifiability. Other removed links belong to downstream signaling, decoupling JNK and IKK from Btk. Furthermore, RPS6 seems to be only regulated by AKT/mTOR and not ERK. HSP27 was not found to be regulated by p38 activity in BL-2 cells. Importantly, by extending the network, the modeling procedure discovered three novel connections that significantly improved the model fit: (i) a positive crosstalk from mTOR to JNK, (ii) a positive feedback from GSK3 to ZAP70 and (iii) a negative crosstalk from p38 on or above RAF/MEK/ERK.

## The network structure learned from BL-2 can be transferred to BL-41 perturbation data

After we have established a semi-quantitative network model of BCR signaling in BL-2 cells, we aimed to assess if the final network describes BCR signaling in a general way. Thus, we performed the same perturbations experiment that was conducted on BL-2 cells on a different BL cell line, BL-41, (**Fig 2A**) and subsequently applied the same modeling strategy (cf. **Fig 1B**) to derive a BL-41 specific signaling network. Starting from the same literature network as in BL-2 cells the network was derived that was best supported by the data. We noted that the reduced chi-square statistics approached the theoretical limit of 1 during network adaptation, indicating accurate model optimization (Fit, **Fig 2B**). Furthermore, the simultaneous independent consistency step (percentage of error reduction compared to unperturbed control) shows that

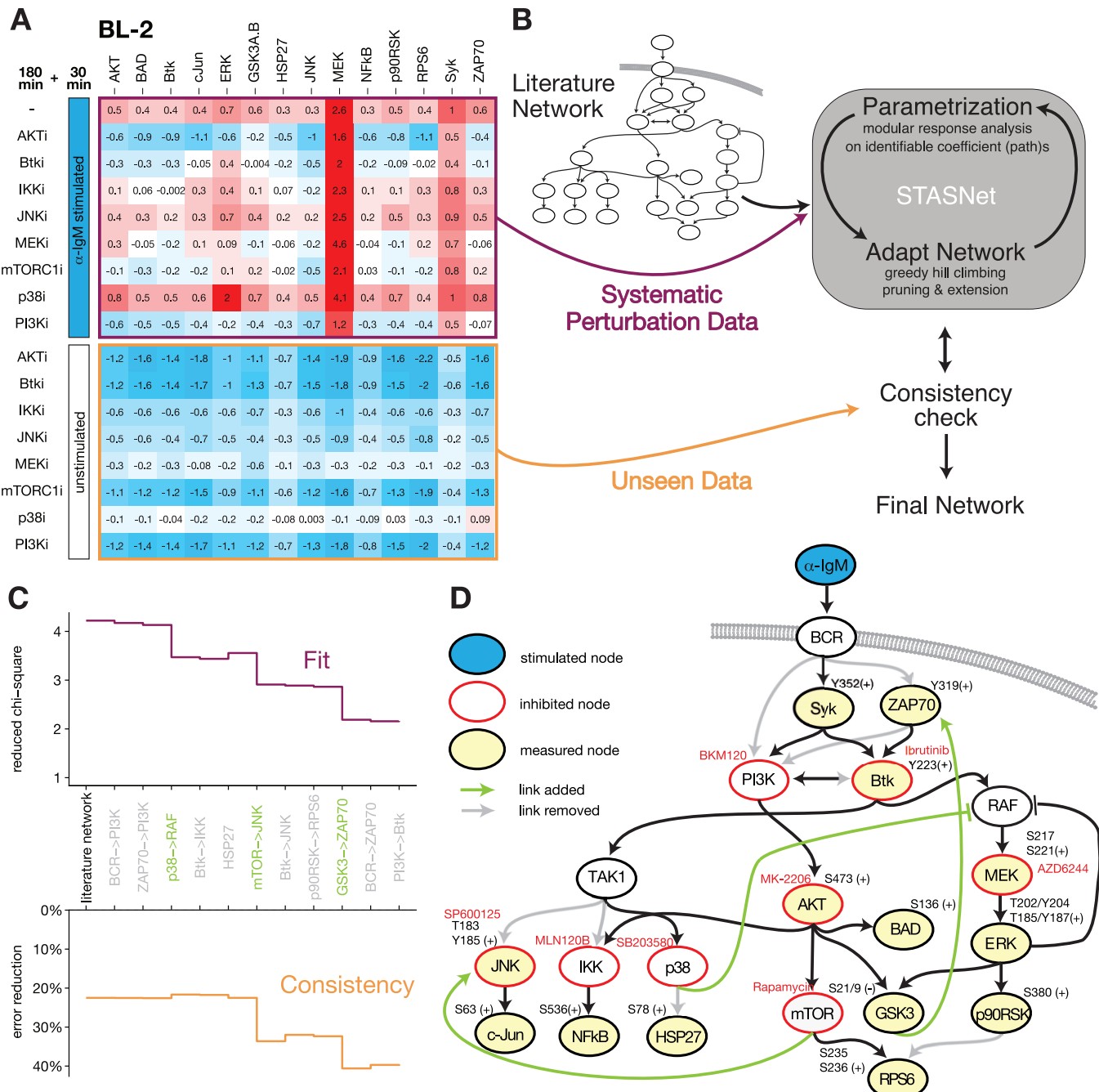

**Fig 1. Discovery of novel feedback and crosstalk structures in a BCR-driven signaling network by perturbation data-based modeling of BL-2 cells.** (**A**) Systematic perturbation data shown as log2 fold changes to solvent control DMSO. Data was generated by pre-treating BL-2 cells with inhibitors targeting key effectors downstream of BCR for 3h with subsequent BCR stimulation using α-IgM for 30 min (upper panel) or no stimulation as consistency check (lower panel). Phosphorylation of indicated signaling proteins (cf. **D**) was measured using bead-based ELISAs (mean, n = 3). (**B**) Modeling workflow using the Modular Response Analysis-based method STASNet to derive a semi-quantitative directed network. The model requires systematic perturbation data (depicted in **A**) and a curated literature network as starting network (cf. **D**). In order to avoid overfitting, the data was split into two parts: (1) α-IgM stimulated data was used for parameter fitting and network adjustment and (2) unstimulated inhibitor data was used for verifying model consistency. After each network adjustment step the unseen data part was simulated and compared. If the error reduction as compared to the null model was not significantly worse, the new network adjustment was upheld otherwise the next best solution was simulated and tested. (**C**) Model performance for each modeling step from the literature-derived starting model to the final model: (TOP) goodness of fit as weighted sum squared residuals divided by number of free parameters and (BOTTOM) consistency check step as percentage of error reduction compared to unperturbed control as null model (see **S1 Text** BL-2_network_model.html: Tab 'Network derivation BL-2'). (**D**) Literature network adjusted to the final signaling network for BL-2 cells derived by the modeling pipeline depicted in **B**. grey line/text—removed links/nodes; green line/text—added links.

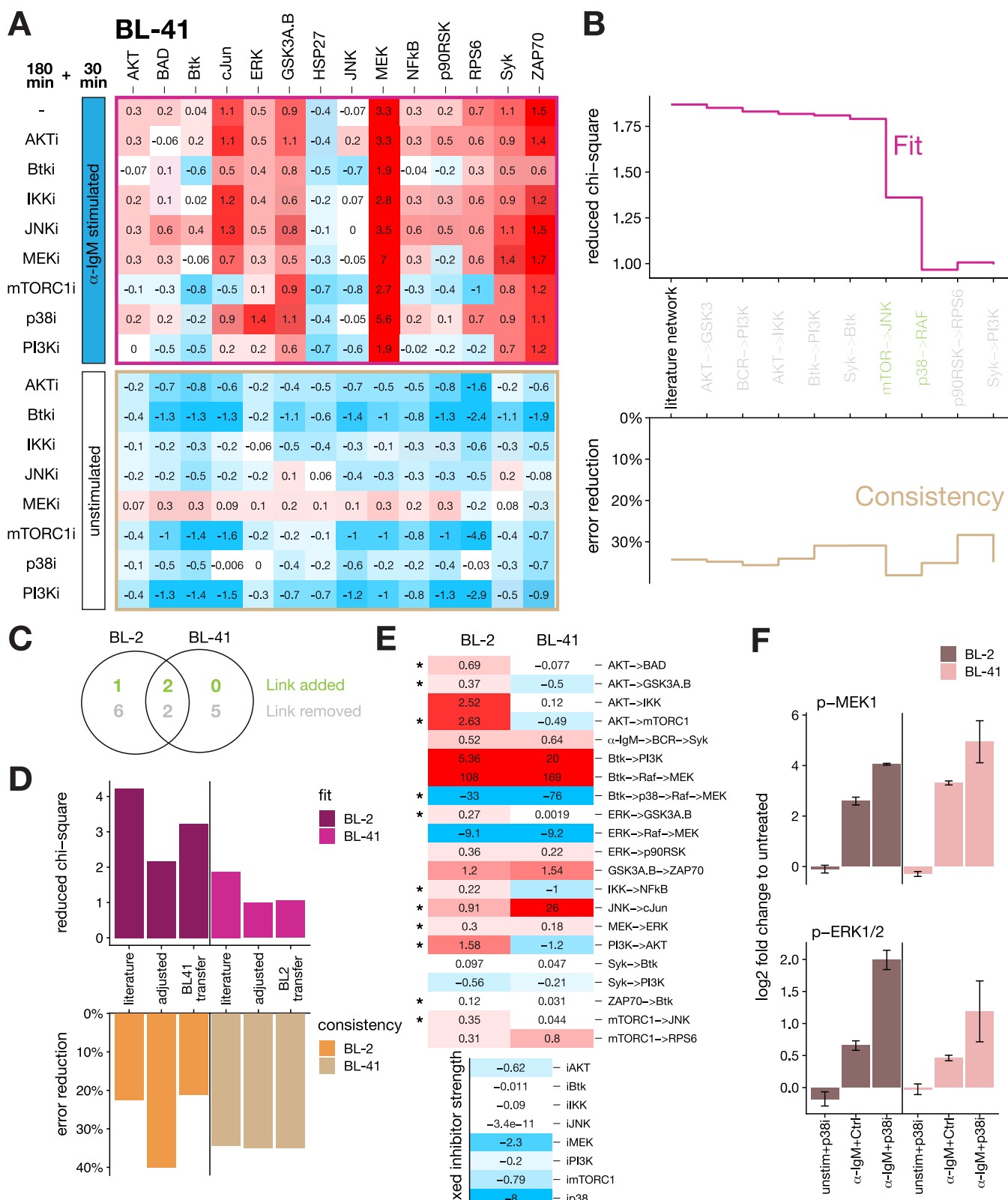

**Fig 2. BL-2-derived modeling structure can be transferred to cell line BL-41.** (**A**) Systematic perturbation data for BL-41 cells generated alike the procedure described in **Fig 1A** (mean, n = 3). (**B**) Model development statistics (TOP) goodness of fit as reduced chi-square and (BOTTOM) unseen data consistency check as percentage of error reduction compared to unperturbed control as null model for each modeling step from the literature-derived starting model (black

and grey arrows in **Fig 1D**) to the final model (grey–reduction, green—extension). See **S1 Text** BL-41_network_model.html: Tab 'Network derivation BL-41'. (**C**) Venn diagram indicating the shared and not shared structural adjustments in the development of BL-2 and BL-41 cells starting from the same literature network (cf. **S3 Fig**). (**D**) Model fit and consistency check statistics for fitted models on BL-2 and BL-41 perturbation data for three different network structures: literature, cell-specific adjusted network (adjusted) and for the best-found structure of the respective other cell line (transfer). See also **S2** and **S4** **Figs**. (**E**) Network coefficients heatmap from models fitted to the BL-2 learned structure for the indicated cell lines. Comparability was ensured by fixing the inhibitor coefficients to BL-2-learned values as both cells received the same inhibitor doses. Stars denote coefficients that are significantly different (i.e., 95%-point wise confidence intervals do not overlap, see **S1 Table**). (**F**) Data excerpt for the model-derived negative crosstalk prediction from p38 to RAF/MEK/ERK pathway in BL-2 and BL-41 cells showing the upregulation of α-IgM-induced activation of pERK and pMEK by the p38 inhibitor SB203580 (mean ± s.e.m., n = 3), but no upregulation by p38 inhibitor alone.

the simulation quality of unseen data stayed approximately at a similar level during network development (Consistency, **Fig 2B**), indicating that no considerable overfitting took place.

During model adaption, 4 adjustments were shared between BL-2 and BL-41. This includes, two of the three additional links already identified in BL-2 cells before, i.e., p38->RAF and mTORC1->JNK. However, 7 and 5 adjustments were BL-2- and BL-41-specific, respectively (**Figs 2C** and **S3**). This divergence in proposed network structure also manifests in the fact that only 14 of the 21 identifiable coefficient (path)s were shared in both cell lines (cf. **S1 Text**, BL-2_network_model.html: Tab '11. Rem. PI3K -> Btk' and BL-41_network_model.html: Tab '9. Rem. Syk -> PI3K'). A structural overlap of only 66% for cell lines from the same lymphoma type would indicate vastly different signaling. However, as our modeling technique employs a greedy hill climbing link adjustment strategy, the resulting networks may represent a local optimum. To further investigate this, we compared goodness of fit and model prediction (consistency) when the respective network structure of one cell line would be used to fit the data of the other cell line. **Fig 2D** shows the best fit and consistency statistics for the transferred networks (transfer) in comparison to the literature-based starting network (literature) and the adjusted literature-based network (adjusted). Interestingly, the signaling network structure learned on BL-2 cells can be faithfully transferred to fit a BL-41 data with strikingly similar statistical properties than the best-found structure found for BL-41 itself. The reverse scenario, i.e., transfer of BL-41 best network structure to BL-2, led to a worse fit and consistency, while only slightly better than the literature network-based model. We assume that the reason for the better transferability is that the BL-2 perturbation data show a higher complexity (above noise) than the BL-41 data, in general and especially for the very central AKT readout. This allows the more complex BL-2 learned network structure to also fit the less complex BL-41 data, but not *vice versa* (compare **S2** and **S4** **Figs**).

Thus, it is evident that the current best network structure found for BL-2 cells can be transferred to BL-41 and results in an equally good model fit as if using the individually fitted network structure from BL-41 data. This successful transfer strongly supports the view of the similarity of BCR signaling in both BL cell lines. Furthermore, it indicates that the BCR signaling network structure developed from BL-2 data can be seen as the most representative BCR signaling structure for BL cells activated by a crosslink of the BCR (**Fig 1C**).

Next, we compared the fitted coefficient (path)s of BL-2 cells and BL-41 cells using the BL-2-optimised network structure (**Fig 2E**). As both cell lines received the same inhibitor dose, the difference in response should be attributed to the network wiring and not to the inhibitor coefficient as the inhibitor coefficients only occur in combination with network coefficient in the model. Therefore, we previously developed the option in STASNet to fix the inhibitor coefficients to a certain value and allow the network coefficient (path)s to adapt accordingly [41]. We decided to fix the inhibitor coefficients to values learned from the BL-2 model so that changes in response are now reflected in the other coefficient path(s). We performed a profile likelihood analysis that unveiled that about half of the non-inhibitor coefficient (path)s (11 of 21, asterisks **Fig 2E**) are significantly different, i.e., the 95% pointwise confidence intervals [47] do not overlap. For example, the coefficient path corresponding to the pathway from Btk via

p38 to MEK is estimated to be stronger in BL-41 than in BL-2 cells. Interestingly, this coefficient path corresponds to the cross talk from p38 to RAF/MEK/ERK, and the underlying data shows that the effect is only notable in the presence of α-IgM stimulus but not when p38 inhibitor is applied without stimulation (**Fig 2F**).

In summary, the BL-2 model constitutes a generic network structure that can accurately describe signaling for both BL cell lines. This finding is important as it demonstrates the existence of a signaling network core for the activated BCR in BL cells. In addition, three hitherto undescribed links in BCR mediated intracellular signaling in BL cells were unveiled to disentangle upstream wiring including the yet not fully resolved receptor proximal signaling events (**Fig 1C**). These so far undescribed network links include the strong negative crosstalk from p38 to RAF/MEK/ERK, which has been previously described in endothelial cells [48] or EGFR-activated epithelial cells [49]. For BCR activated signaling this feedback however was not yet described. Therefore, we decided to conduct further experiments to characterize the model-predicted p38-mediated dampening of MEK/ERK phosphorylation.

### An α-IgM dependent feedback signal from p38 (MAPK14) attenuates pMEK and pERK

Since the model-based findings rely on a single time point (30min after α-IgM treatment), we investigated the temporal dynamics of MEK and ERK phosphorylation in BL-2 cells (**Fig 3A**). An increase in MEK and ERK phosphorylation as early as 2 min after α-IgM-mediated BCR stimulation is observed in the presence of the p38 inhibitor SB203580. This p38 inhibitor-dependent increase of MEK/ERK phosphorylation persists for the entire span of the measured 60 minutes post stimulation with the strongest induction within the first 10 minutes and a slight decrease thereafter (**Fig 3A**).

To directly asses the involvement of p38 in this predicted cross talk, a knockdown of p38α (MAPK14) was performed. p38α is one of the most abundantly expressed isoforms of p38 in BL-2 cells (**S5 Fig**). Importantly, the knockdown of p38α affects the phosphorylation of ERK similarly to the previous p38 inhibition (**Fig 3B**). This further supports the observation that p38 activity dampens the MEK/ERK pathway and that in BL-2 cells this seems to be conferred by the α-isoform of p38.

By analyzing subcellularly fractionated cell lysates, it can be shown that both cytosolic as well as nuclear pERK is increased in the presence of the p38 inhibitor SB203580 in α-IgM-stimulated BL-2 cells (**Fig 3C**). As only active ERK is able to enter the nucleus but not MEK [50] it is unlikely that the cause of the upregulation is a prolonged retention of ERK together with active MEK in the cytosolic scaffold. This finding also indicates that ERKs activity is likely to be propagated to its many nuclear targets even in the presence of p38 inhibitors.

Having verified the negative crosstalk from p38 on MEK and ERK, the next step was to characterize the molecular mechanism that underlies this crosstalk. The RAF-MEK-ERK signaling cascade is a well-characterized MAPK pathway involved in different cellular processes initiated by different cell surface receptors and RAS. Upstream kinase RAF is known to be negatively regulated by phosphorylation at serines 289, 296 and 301, which are known negative feedback sites phosphorylated by ERK [51]. Since p38, as ERK, belongs to the family of MAPK we examined whether p38 regulates phosphorylation at serine 289, 296 and 301 and thus might affect RAF1 activity. After 30 minutes of α-IgM stimulation, RAF1$^{S289/296/301}$ phosphorylation is clearly detectable and diminished by MEK inhibition using AZD6244, confirming the regulation by ERK also within the BCR signaling cascade (**Fig 3D**). However, RAF1 phosphorylation at these sites is not affected by p38 inhibitor SB203580 (**Fig 3D**). Also, in BL-2 and CA-46 Burkitt lymphoma cells no difference on the RAF1 feedback sites can be seen (**Fig 3E**).

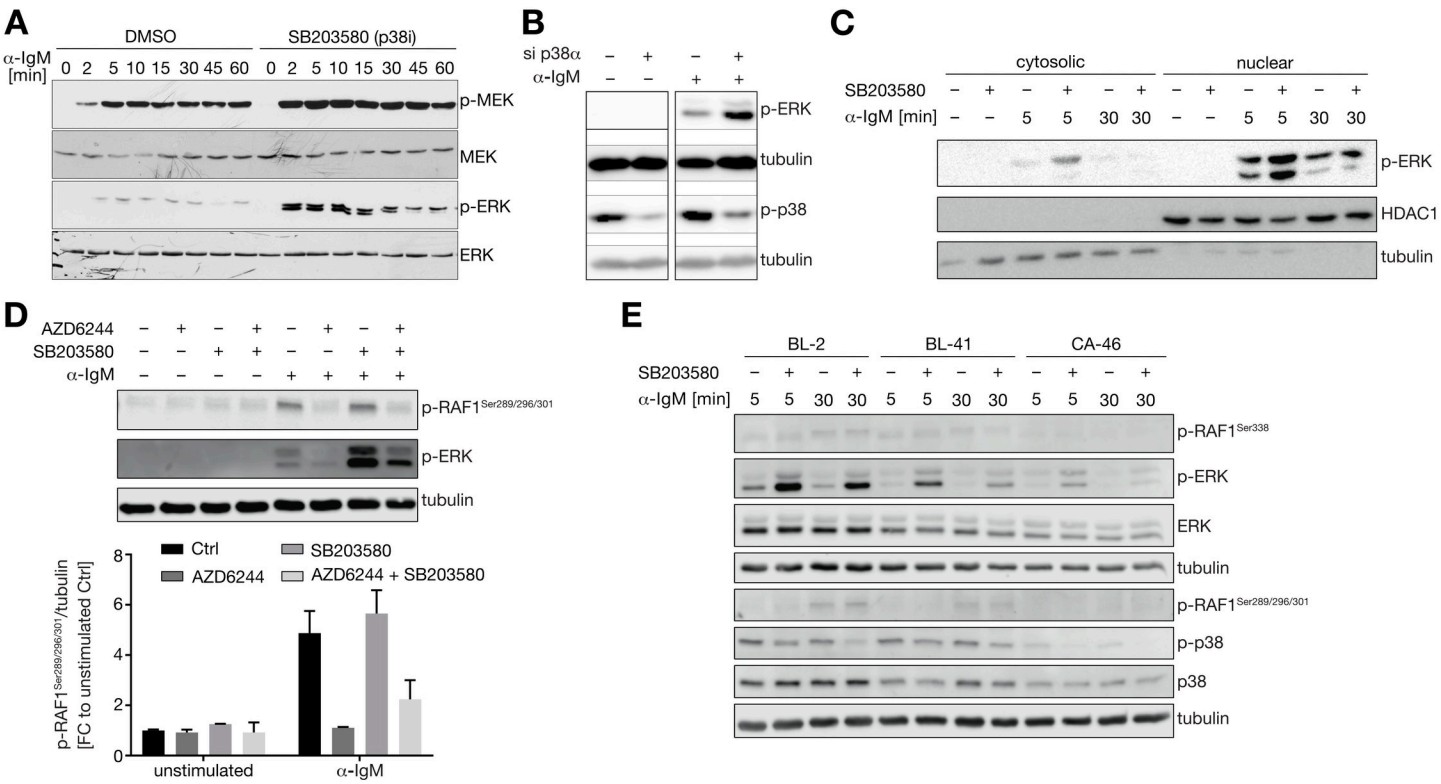

**Fig 3. Increased MEK/ERK-pathway activity in BCR-activated B cell lines after p38 intervention.** (**A**) Changes in the phosphorylation of MEK and ERK in BL-2 cells after treatment with α-IgM in the presence or absence of the p38 inhibitor SB203580. (**B**) Phosphorylation of ERK is further increased in α-IgM-treated BL-2 cells after 24h of p38α (MAPK14) knockdown. (**C**) Phosphorylation of ERK is enriched within the nucleus of α-IgM treated BL-2 which is further enhanced by inhibiting p38. Tubulin and HDAC1 were used as reference for the cytosolic and nuclear fraction, respectively. (**D**) (TOP) Phosphorylation of c-RAF at serine-residues 289/296/301 is increased after 30 min α-IgM treatment in BL-41 cells but not affected by p38 inhibition. The inhibition of MEK using AZD6244 interrupts the phosphorylation of RAF. Representative Western blot. (BOTTOM) Bar plots quantifying c-RAF phosphorylation measurements for n = 2 replicates. (**E**) p38 affects the MEK/ERK pathway in a comparable way in different BL cell lines after α-IgM treatment. Shown are phosphorylations of Raf1 at serine 338 (activatory site) and 289/296/301 (ERK feedback sites) as well as of ERK and p38 in Burkitt lymphoma cells BL-2, BL-41 and CA-46.

This indicates that p38 must attenuate the MEK-ERK pathway directly after BCR activation while the feedback from ERK to RAF1 seems p38-independent.

Next to the feedback we also checked upstream activation of RAF1 by measuring phosphorylation at serine 388 [52] in three different BL cell lines BL-2, BL-41 and CA-46 (**Fig 3E**). Phosphorylation of S338 increases slightly after BCR activation, however, there is no effect by p38 inhibition. Importantly, in all three cell lines p38 inhibition affects the MEK/ERK pathway in a comparable way after α-IgM treatment as shown by upregulation of pERK (**Fig 3E**). Therefore, our data support the view that this crosstalk is a widely existing phenomenon in BL cells. In addition, we also saw that CD40L stimulation in CA-46 cells could also produce hyperactivation of pERK together with p38 inhibition (**S6 Fig**). This demonstrates that the observed p38-ERK crosstalk is not only limited to BCR activation, but is an inherent mechanism in BL signaling.

## Global phosphoproteomic analysis of BL-2 cells supports model-derived pathway network crosstalk

To further characterize the crosstalk from p38 to MEK/ERK and to investigate more general consequences on overall signaling, a systematic proteomic analysis was performed.

Specifically, we performed phosphoproteomic mass spectrometry combined with the tandem mass tag (TMT) technology that allows to quantitatively compare phosphorylation in different samples without dropouts [53].

Phosphoproteomes were obtained for unstimulated controls, and cells stimulated for 30 min with α-IgM that were incubated with inhibitors of p38 (SB203580), PI3K (BKM120) and mTORC1 (Rapamycin) or solvent control (DMSO) as in the above-described analysis. PI3K inhibitor was chosen as it represents a central hub in the deduced network model with mTORC1 the target of rapamycin being a major downstream mediator, whereas the p38 inhibition was included to get an unbiased insight into the downstream effects of p38 in BCR signaling as well as to extend the understanding of the p38-MEK/ERK crosstalk in B cells.

Overall, 28871 phosphosites from 5698 proteins (localization probability>0.75) were identified reliably. When investigating the global effect on the 3000 most varying phosphosites by hierarchical clustering (**Fig 4A**) and principal component analysis (**Fig 4B**), it is evident that α-IgM treatment had the strongest impact. Interestingly, inhibition of PI3K by BKM120 reverted the effect of α-IgM partially, and thus α-IgM+PI3Ki samples were located between Control+DMSO and α-IgM+DMSO samples in PC1-PC2 space. α-IgM treatment in combination with Rapamycin (mTORC1) or SB203580 (p38) clustered and colocalized well with α-IgM+DMSO treatment. This indicates that the inhibition of mTORC1 and p38 had a rather confined influence on the phosphoproteome in α-IgM activated BL-2 cells.

Differential expression analysis showed that α-IgM treatment had the largest effect with 8094 phosphosites (28% of detectable phosphosites) that were significantly different compared to untreated control (limma FDR≤5%, **Fig 4C**). PI3K inhibition had an influence on 1433 phosphosites when compared to α-IgM treatment alone. mTORC1 inhibition led to a significant change in only 259 phosphosites, of which 47% are shared sites with its upstream regulator PI3K. With 12%, a considerable part of the α-IgM-regulated phosphoproteome is counter-regulated by PI3K (980/8094). Interestingly this PI3Ki counter-regulation is much more prominent for α-IgM down-regulated sites (23%) than for α-IgM upregulated sites (3%). Due to its large counter-active potential, it can be assumed that PI3K is a very potent target for treating BCR-addicted neoplasms.

Differences in substrate phosphorylation are usually caused by differences in kinase activity. Therefore, upstream kinase annotation provided by the PhosphoSitePlus Database [54] was used to get an insight into the potential kinase activity by a combined analysis of their target sites (**Fig 4D**). Significant kinase activity changes were defined by testing all treatments to α-IgM treatment alone using a pairwise t-test. Kinases acting further upstream such as Syk and ZAP70 are most strongly regulated by α-IgM vs Control treatment. The effect of inhibitors on those upstream kinases was rather small with marginally significant effects of PI3Ki counter-acting SYK-target induction and mTORi increasing ZAP70 target induction (**Fig 4D**). Reassuringly, *bona fide* target kinases for the inhibitors, i.e., AKT for PI3K inhibition and mTOR/p70S6K for mTOR inhibition showed the strongest and most significant reversal of α-IgM induction. Importantly, the proposed p38-MEK/ERK crosstalk is also visible in the significant upregulation of MEK and ERK target site phosphorylations after SB203580 treatment. This further indicates that the crosstalk is functional and inhibition of p38 leads to an enhanced ERK signaling output on a global scale.

Having ascertained the negative p38-MEK/ERK crosstalk, the significantly regulated sites (limma, FDR≤5%) by SB203580 were further investigated (**Fig 4E**). p38 inhibitor responsive phosphosites are rather specifically regulated, with 70% (24/34) of significantly regulated sites neither co-regulated by other inhibitors nor counteractively regulated by α-IgM treatment. 12 sites are strongly downregulated and 22 hyperactivated compared to Control and DMSO treatment. Three of the hyperactivated sites are activatory phosphosites of ERK1/2 (red)

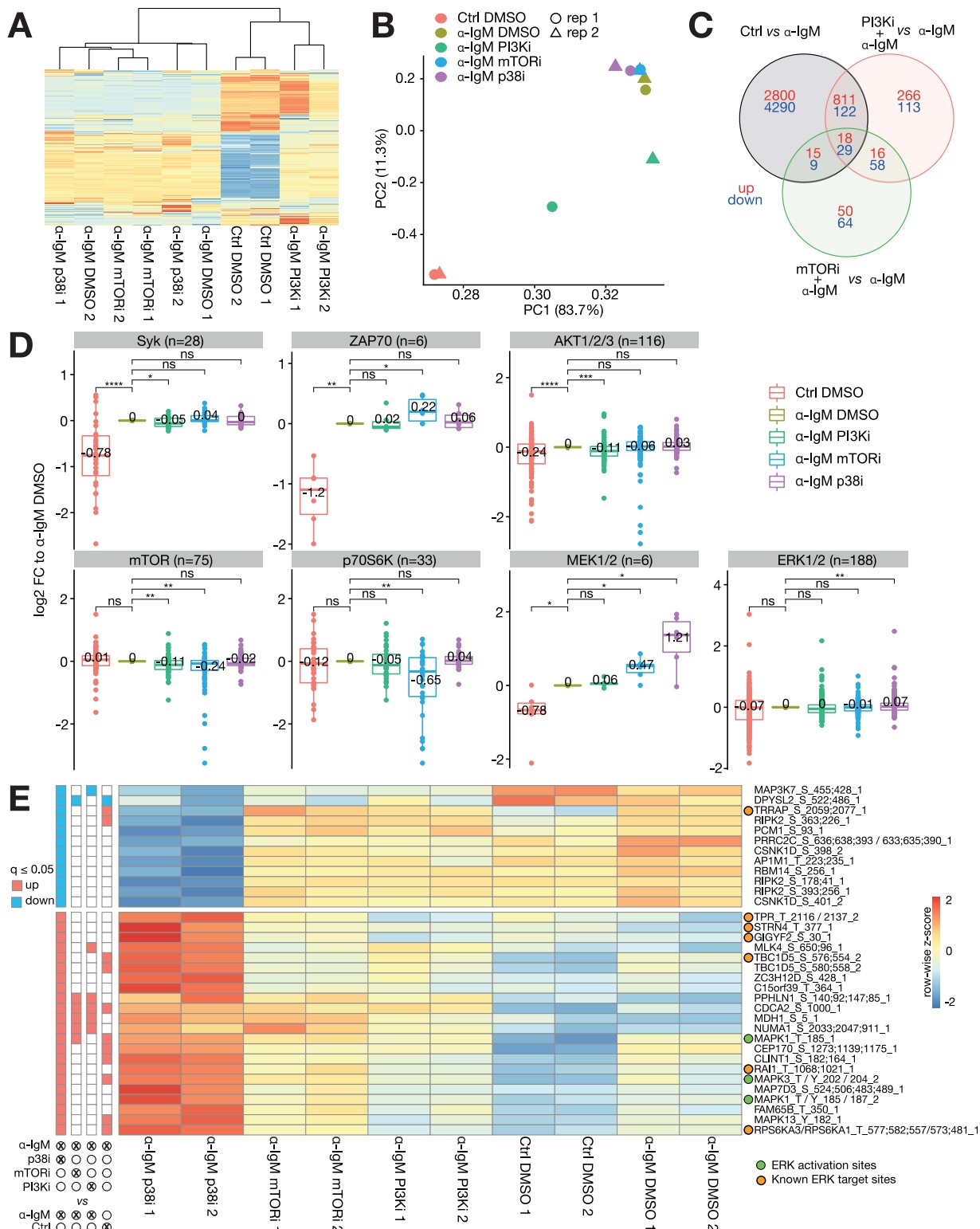

**Fig 4. Phosphoproteomics analysis supports the BCR-signaling model and reveals a dominant effect of the PI3K pathway inhibition onto BCR-signaling in BL-2 cells.** Analysis of Tandem-Mass-Tag (TMT) Mass spectrometry measurements for BL-2 cells treated with α-IgM and inhibitor solvent DMSO or inhibitors of PI3K (BKM120), MTORC1 (Rapamycin) or p38 (SB203580) (n = 2). (**A**) Hierarchical clustering of 3000 most varying phosphosites demonstrates a global effect of α-IgM and PI3K inhibitor BKM120 on the phosphoproteome and subtle effects of the remaining inhibitors. (**B**) Principal component analysis shows that α-IgM effect is governing the principal components 1 and 2. Only

PI3Ki treatment is able to partly revert the α-IgM effect. (**C**) Overlap of differentially regulated phosphosites (limma, FDR≤5%) for indicated selected comparisons. (**D**) Upstream kinase activity assessment on base of log2 fold changes (vs. α-IgM+DMSO) in PhosphoSitePlus-annotated target sites (Nov 2021) for selected kinases. Significance asserted by two-sided t-test: ns—not significant; * - 0.05; ** - 0.01; *** - 0.001; **** - 0.0001); Average value indicated. (**E**). Phosphosites significantly regulated by p38 inhibitor SB203580 (limma, FDR≤5%). Left panel denotes which site was found to be significantly regulated (blue—down; red—up) by the indicated comparison. Sites are annotated as follows: '*HGNC symbol*'_'*amino acid*'_'*position*'_'*number. of phosphosites*'; ERK activation sites and known target sites of ERK [55] are indicated by green and orange circles, respectively.

demonstrating a direct ERK hyperactivation. We then checked other phosphosites for known ERK-targets, using a previously published compendium of ERK targets [55] Seven of the p38 inhibitor-dependent phosphosites are described as ERK targets, of which six follow the same hyperactivation pattern as ERK phosphorylation itself. This further corroborates that next to ERK also downstream partners are hyperactivated by p38 inhibition. Notably TRRAP, the only identified known ERK target to be downregulated by p38 inhibition seems to be a shared target of both ERK and p38. This can be proposed based on a SILAC-based phosphoproteomics study which observed, that TRRAP$^{S2077}$ is upregulated after 15 min EGF stimulation and downregulated by EGF stimulation and p38 inhibitor (SB202190) treatment in HeLa cells [56]. Taking into account the similarity of the results in that study and our own observation we conclude that TRRAP$^{S2077}$ is dominated by p38 over ERK activity.

From the phosphoproteome analysis, we conclude that p38 inhibition indeed leads to stronger ERK activation, but it is rather confined to a small subset of the signaling network. While the mechanism of the crosstalk remains unclear, the phosphoproteome does identify 19 candidate phosphosites that are p38-regulated but not known ERK targets which could be investigated in future studies.

## BL-2-dervied network improves starting basis for network development of DLBCLs

Having developed a semi-quantitative model for activated BCR signaling in BL cells we sought to investigate if a similar signaling network exists in cells with aberrant BCR signaling such as in ABC-like DLBCL. For that reason, the intracellular signaling networks of two representative cell lines HBL-1 and OCI-LY3 were analyzed. Similarly, to BL-cell lines earlier (cf. **Fig 1A**), the signaling in both cell lines was perturbed using the same inhibition and measuring scheme. Additional α-IgM stimulation was not required, as both cell lines exhibit chronically active BCR signaling [9]. The resulting phosphoprotein perturbation data of both cell lines (**Fig 5A**) were analyzed using our STASNet pipeline.

We first tested if the literature-based network or the BL-2-derived network provides a better starting point for the development of the signaling model. We trained models for both cell lines on (a) the literature network (cf. **Fig 1D** grey and black arrows) and (b) BL-2-derived final network (cf. **Fig 1D**) without receptor proximal signaling as no stimulation perturbation was present. For both DLBCL cell lines, we found that the BL-2-derived model resulted in a better fit compared to the literature-based model (**Fig 5B**). We then structurally adjusted the BL2-network to the respective DLBCL-cell line reaching reduced chi-square residuals close to the theoretical expectance for optimal fit, i.e., 1 (**Fig 5B**–adjusted from BL-2). We also tested whether those cell line-specific fits yield generality in DLBCLs and re-fitted HBL-1 and OCI-LY3 data with the best network structure identified for the respective other cell line. Since both cross-fittings resulted in clearly higher residuals than the respective self-best network fit, we decided to use the individually developed networks for the further analysis.

The corresponding network starting from the BL-2 model (without upstream wiring due to missing perturbations above PI3K and Btk) is presented in **Fig 5C** before and after training on

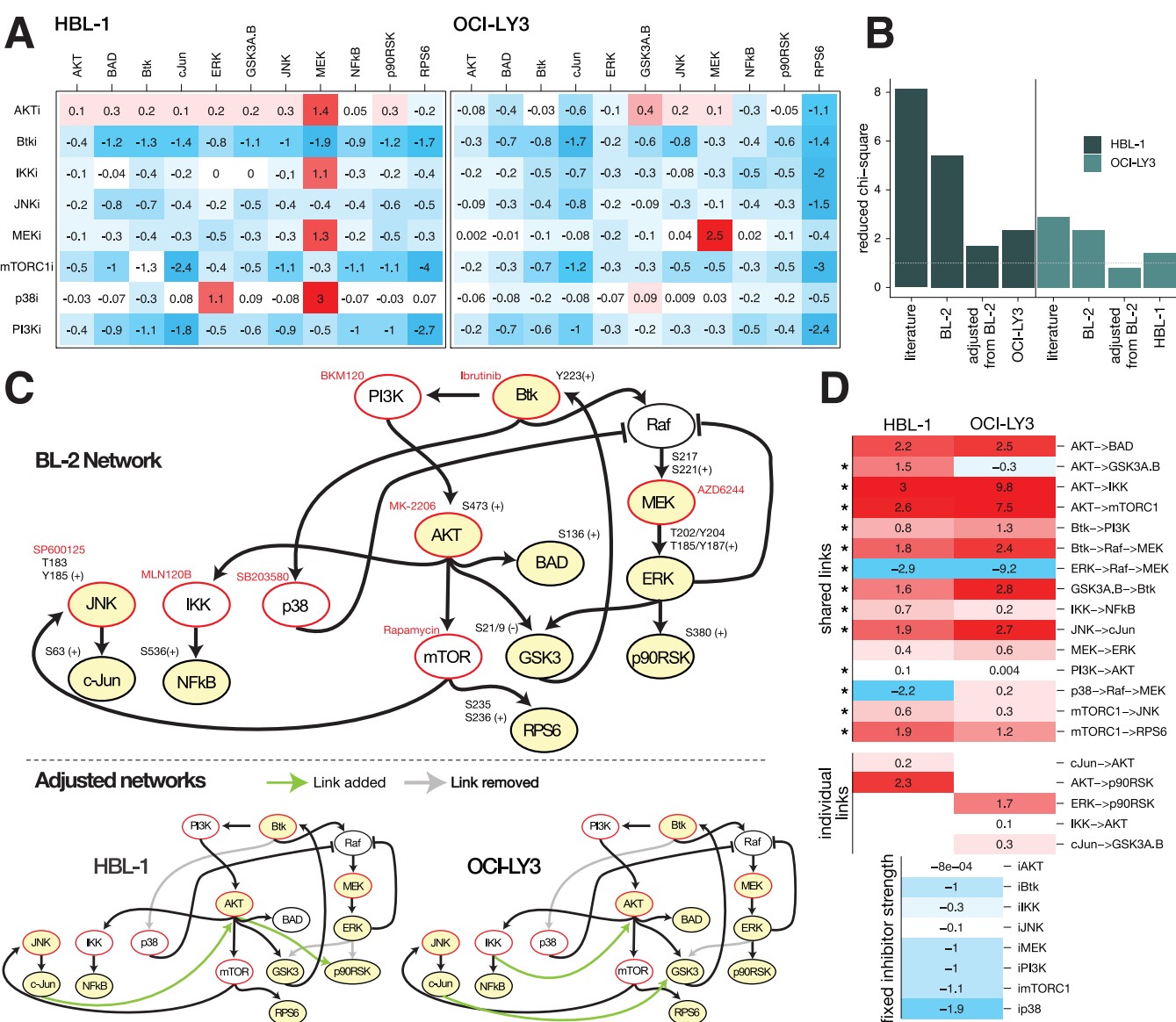

**Fig 5. BL-2-derived network structure sets a veritable starting base to develop networks for DLBCL cell lines HBL-1 and OCI-LY3.** (**A**) Systematic perturbation data of bead-based ELISA measurements of the DLBCL cell lines HBL-1 and OCI-LY3 quantified as log2 fold changes to solvent control (DMSO); mean of n = 3. (**B**) Goodness of fit expressed as reduced chi-square statistic $X_r$ on selected network structures for the two DLBCL cell lines HBL-1 and OCI-LY3. *literature*–network from **Fig 1B**; *BL-2* –BL-2 network derived from **Fig 1C**; *adjusted from BL-2* –BL-2 network locally adjusted to respective DLBCL cell line; *OCI-LY3/HBL-1* –final adjusted network of respective other DLBCL cell line (see **C**). (**C**) Network structures of BL-2 derived starting network and final DLBC-specific networks trained on HBL-1 and OCI-LY3 data (see **A**). (**D**) Side-by-side comparison of the model coefficient (path)s (log scale) with fixed inhibitor strengths set to mean of both cell line models. Empty tiles indicate missing links in one of the cell lines (individual links). Asterisks point to non-overlapping confidence intervals as estimated by STASNet profile likelihood function (see **S2 Table**).

HBL-1 and OCI-LY3 cell lines. In both cell lines the links from Btk to p38 and ERK to GSK3 were removed by the network adjustment strategy of STASNet (**Fig 5C**). In HBL-1 cells additional links from JNK/cJun to AKT and from AKT to p90RSK were required, whereas in OCI-LY3 the network required a link between JNK/cJun and GSK3 and between IKK and AKT for optimal fits. The link from ERK to p90RSK was removed only in the HBL-1 network. In agreement with the findings from the analysis of the Burkitt lymphoma cell line BL-2, we also note a central role of PI3K in the studied DLBCLs indicated by the network central

position of the PI3K node even after the individual network adjustment (**Fig 5C, bottom**), i.e., every network node can be reached by PI3K. In addition, when inspecting the links removed and added in both DLBCL cell lines, a shift from less control by MAPK to more control by mTOR/AKT can be noticed which further corroborates the importance of PI3K in DLBCL signaling. This PI3K/mTOR-dependency is further underlined by the growth inhibitory effect of corresponding inhibitors in those cell lines (see **S7 Fig**).

Next to identifying the structural similarities and differences, STASNet allows to reveal the quantitative signaling differences in the cell lines by comparing the coefficient (path)s (**Fig 5D**). For better comparability we fixed the inhibitor coefficients to the mean of both cell line models. After applying parameter stability analysis using profile likelihood, we noted that of the 15 shared links only 2 (AKT->BAD, MEK->ERK) are not significantly different (i.e., 95% confidence intervals do not overlap; **S2 Table**). Of note the p38->Raf->MEK crosstalk is negative in HBL-1, like in BL-2 and BL-41, but faintly positive in OCI-LY3.

We found that from 18 identifiable coefficient (path)s that are present in the BL-2 network model aside from receptor-proximal signaling (**Fig 5C TOP**), 15 (83%) and 16 (89%) coefficient (path)s were also required in HBL-1 and OCI-LY3 cell line models, respectively. Next to the removal of three (HBL-1) and two (OCI-LY3) links, both models required only two additional coefficients for their final best fit, suggesting that many insights from α-IgM-stimulated BL cells can be transferred to DLBCLs. However, these changes are necessary, as we observe a ~3-fold improvement of fit, i.e., reduction of $X_r$, in both DLBCL cell lines with five and four adjustments for HBL-1 and OCI-LY3, respectively (**Fig 5B**). This indicates that a common model of BL and DLBCL would perform poorly and local adjustments are still required but that a common conserved core network of BCR downstream signaling exists.

## Conserved core network of chronic and acute B cell receptor signaling

To further analyze similarities and differences between the modelled cell lines, we decided to investigate the core network. We defined the core network to consist of all edges that are present in at least three of the four final cell line models, which produces a network of 16 nodes and 16 edges (**Fig 6**). For better comparability we binned the coefficients into five distinct states: amplification (r > 1), dampening to neutral relay (0 < r < 1), no link (r = 0), attenuating to neutral inhibition (0 > r > -1) and enforced inhibition (r < -1).

When analyzing the overlap, we find that 9 of 16 coefficients are qualitatively similar, i.e., coefficients are either all positive or all negative. This number can be elevated to 12 (75%) by taking into account that the coefficients are multiplied along the path such that a negative qualitative response coefficient between PI3K and AKT in BL-41 cells leads to an overall positive response from PI3K to AKT downstream targets mTOR, BAD, NFkB and GSK3. This renders the sign of the response same for three of the AKT downstream targets in all four cell line models when viewed from PI3K. The negative quantification of the PI3K-> AKT link in BL-41 cells can be attributed to the weak AKT signal found in that cell line (cf. **Fig 2A**). When using the above-mentioned interpretation for BL-41 coefficients downstream of AKT, we note that for all 16 coefficients at least three out of four exhibit the same sign, indicating a strong overall signaling conservation. The deviating coefficients for one cell line model are the aforementioned negative impact of PI3K on AKT in BL-41 cells, a negative effect from AKT to GSK3 and a weakly positive p38->RAF crosstalk in OCI-LY3 (whose confidence interval spans over 0 indicating unimportance of that link, cf. **S2 Table**) and a missing link in ERK-> p90RSK in HBL-1 cells. Since none of these outlier links is inherent to both cell lines of DLBCL or Burkitt Lymphoma origin we are unable to assign those differences to adaptations to chronic B-cell receptor signaling. Looking for more subtle common changes between acute and chronic BCR

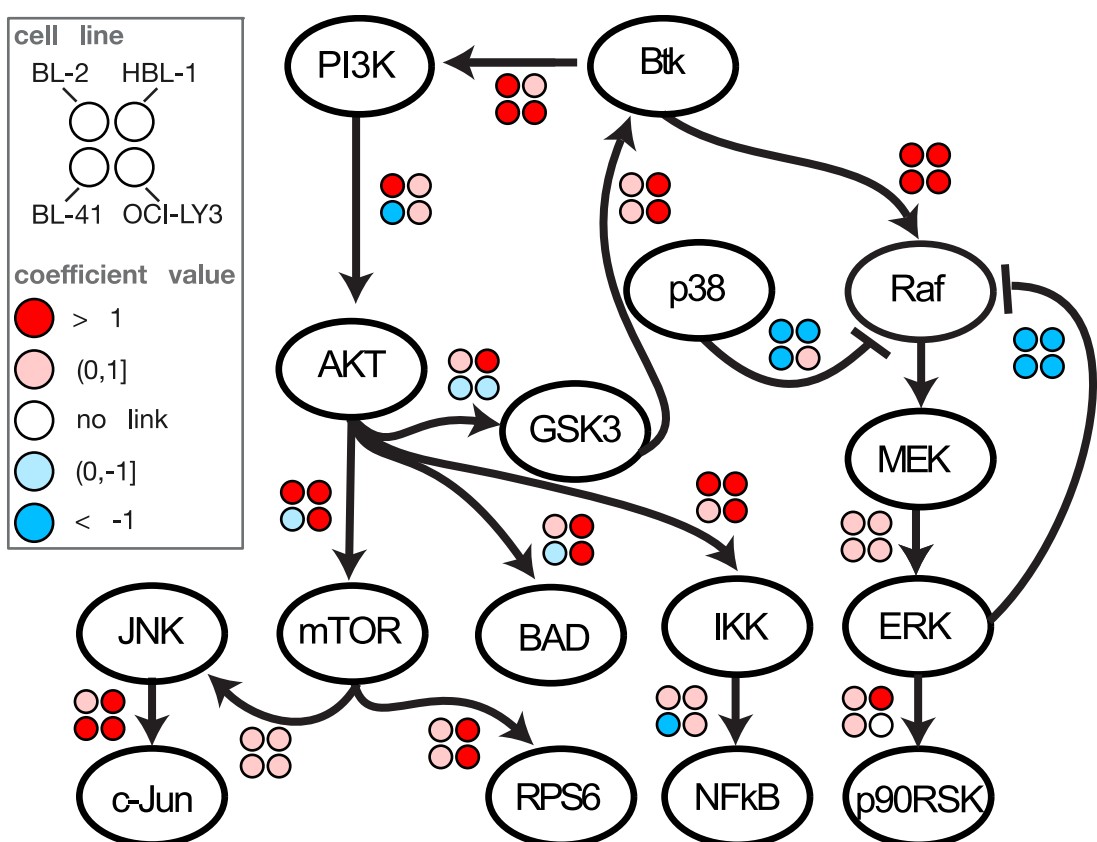

**Fig 6. Conserved core network of chronic and acute B cell receptor signaling.** Consensus network structure and coefficient quantification of links present in at least 3 of the 4 modelled cell lines with chronic (HBl-1, OCI-LY3) and acute (BL-2, BL-41) BCR signaling. BL-2 and BL-41 information retained from models trained on the BL-2 structure and with inhibitor strength fixed to BL-2 model (**Fig 2E** and **S1 Table**), the connection for GSK3 to Btk was retrieved by multiplying the coefficients r_ZAP70_GSK3A.B with r_Btk_ZAP70. HBL-1 and OCI-LY3 information retained from their respective adjusted model when started from the BL-2 network with inhibitor strength fixed to average of both cell line models. (**Fig 5D** and **S2 Table**). Note that the coefficients for ERK->RAF, p38-> RAF and Btk->RAF all occur in combination with the RAF->MEK coefficient so that the modelled strength represents the response on MEK instead of RAF.

signaling a stronger positive feedback from GSK3 and a stronger coefficient between mTOR and RPS6 for HBL-1 and OCI-LY3 is observed. This points to the fact that only subtle changes seem to occur. In summary, we quantitatively describe a conserved core network in the studied cell systems.

## Conclusion

In this work, we established a (semi)-quantitative signaling network of the BCR and tested it in different B cell lymphoma cell lines. We explored how a more quantitative understanding of signaling can be used to describe a conserved B cell receptor regulated signaling network, which can be used as a starting point to quantify cell-specific, individual networks. The development of the new BCR network model allowed us to introduce so far not described links in BCR mediated intracellular signaling. Earlier described network cross talks and feedbacks were confirmed, but also new feedbacks are identified further supporting the complexity of the BCR signaling network. Thus, the well-known RAF->MEK->ERK-|RAF regulatory loop also exists in BLs and DLBCLs, whereas the negative impact of BCR mediated p38 activation onto this loop is a newly discovered crosstalk found in three BL cell lines BL-2, BL-41 and CA-46

and one DLBCL cell line HBL-1. We propose that the identified negative crosstalk from BCR mediated activation of p38 to the RAF/MEK/ERK pathway might dampen an excessive RAF/MEK/ERK signaling cascade. Despite the discovery of feedbacks and crosstalks in ERK and p38 signaling we could not find a cellular dependency in the cell lines we studied. However, as we determined that those signaling motifs are conserved across different B cell lymphoma types it might be possible that insights of this study apply also to other B cell lymphomas. For example, a recent study described an essential dependency to MAPK activity in Chronic Lymphocytic Leukemia progression suggesting that insights from this study could be valuable for treatment decisions in CLL [57].

While the mechanism behind the p38 –MEK/ERK crosstalk remains enigmatic, it is clear that it is neither mediated by the ERK feedback sites nor the tested activation sites of RAF1. RAF is a complex kinase family with more activity regulating phosphosites than we could observe with our analysis and at least another important isoform BRAF [58]. Therefore, we cannot rule out that the effect could still manifest on other RAF1 activity sites or the BRAF isoform. As RAF activity is neither measured nor perturbed, the model coefficients of the p38 crosstalk are only identifiable as combinations with the RAF->MEK link (**S1** and **S2 Tables**) which means a direct crosstalk from p38 to MEK would result in the same fit. A previous study on Hela cells determined that the crosstalk is conferred by direct binding of p38 to ERK [59], as it demonstrated direct binding and no change in MEK phosphorylation. In our study, as all cell line models retained a strong negative feedback from ERK to RAF (**Fig 6**)—which causes an hyperactivation of MEK, when ERK activity is switched off by e.g., a MEK inhibitor [51, 60]—we would expect a downregulation of MEK activity when ERK is hyperactivated alone. Since in our data instead of an attenuation we observe an upregulation of MEK phosphorylation in cells that show the p38 crosstalk when treated with p38 inhibitor (**Figs 2F**, **3A** and **5A**), we postulate that the crosstalk must act on or upstream of MEK.

In addition, a positive link between mTOR and JNK and a positive feedback downstream of AKT to upstream of PI3K/Btk is proposed in our consensus model, which notably are kept also in both DLBCL cell line models. Importantly, also the qualitative wiring of the established core BCR network is conserved across all or at least the majority of cell lines and only few cell type- and cell line-specific characteristics could be detected.

In conclusion, we could demonstrate that the more generic network trained from BL-2 and BL-41 cell lines had better transfer capacity to DLBCL cell lines HBL-1 and OCILY3 than the literature network indicating that a model trained on BCR-activated B-cells is informative to be transferred to BCR-driven cancer models. Importantly, our conserved BCR network model can be applied to the situation in antigen-driven BCR activation (BL-2/BL-41) and the situation, were the BCR pathway is antigen independently activated, driven by gain-of-function mutations in specific signaling molecules of the BCR pathway (HBL-1/OCI-LY3). A recent study that compared primary Chronic Lymphocytic Leukemia and non-malignant B Cell signaling by modeling a smaller network (5 nodes) using Bayesian inference learning underscores our finding by determining only a modest topology difference between the malignant and non-malignant network [61]. In here we can recapitulate this finding on a larger core network (16 nodes, 16 edges) for aberrant and non-aberrant B-cell receptor driven signaling (**Fig 6**).

## Material and Methods

### Cell culture and stimulation

BL-2, BL-41, and OCI-Ly3 cells were obtained from the DSMZ (Braunschweig, Germany). HBL-1 cells were kindly provided by D. Krappmann (Munich, Germany). BL-2 and BL-41 cells were cultivated as described previously at cell densities between $2x10^5$ and $1x10^6$ cells/ml

[62]. For stimulation studies, BL-2 and BL-41 cells were cultured in RPMI1640 with 10% FCS at $3 \times 10^5$ cells/ml and incubated with 1.3μg/ml goat α-IgM F(ab)$_2$ fragments (Jackson Immunity) for indicated time points. For pathway interventions cells were treated with DMSO, 5μM SP600125, (all Merck KGaA, Darmstadt, DE), 1μM AZD6244, 2μM BKM120, 10μM Ibrutinib, 1μM MK-2206, 1μM Rapamycin (all Selleckchem, Munich, DE), 10μM MLN120B (MedChemExpress, Sollentuna, SE), 2μM SB203580 (Sigma-Aldrich, St. Louis, US). HBL-1 or OCI-LY3 cells were cultured in RPMI1640 with 10% FCS at $5 \times 10^5$ cells/ml. For studying protein phosphorylation, the cells were incubated with inhibitors for 3 hours. HBL-1 and OCI-LY3 were then harvested, while BL-2 and BL-41 cells were incubated for additional 30 minutes with 1.3μg/ml goat α-IgM F(ab)$_2$ fragments (BCR activation) or left untreated. Cells were harvested using corresponding inhibitors of phosphatases (PhosSTOP Roche, Basel, CH) and proteases (cOmplete Mini Roche, Basel, CH) and protein was isolated using RIPA buffer.

## Western blot analysis

Cells were analysed for protein expression by SDS polyacrylamide gel electrophoresis and Western blot analysis using the following antibodies: mouse monoclonal anti α-tubulin (#05–829, Merck Millipore, Burlington, US), rabbit α-p-AKT$^{Ser473}$ (#9271), rabbit α-pan AKT (#9272), rabbit α-p-p38 (#9211), rabbit α-p38 (#9212), rabbit α-p42/44 (#4695), mouse α-MEK1/2 (#4694), rabbit α-p-p42/44 (#4370), rabbit α-p-p42/44 (#4377), rabbit α-p-MEK1/2 (#9154), rabbit α-p70 S6 Kinase (49D7) (#5707), rabbit α-p-p70 S6 Kinase$^{Thr389}$ (108D2) (#9234), α-p-Raf1$^{Ser289/296/301}$(#9431), rabbit α-p-Raf1$^{Ser338}$ (56A6) (#9427)(all from Cell Signaling Technology) and α-mouse HRP polyclonal goat (D1609) and α-rabbit HRP polyclonal goat (E1710) (all from Santa Cruz Biotechnology, Inc.).

## Bio-Plex multiplex immunoassay

To measure all protein phosphorylations in one sample, the magnetic bead-based multiplex assay (BIO-RAD) was performed. The principle is that a specific antibody coupled to a color-coded bead identifies the total protein of interest while a second detection antibody determines the magnitude of a distinct phosphorylation. For this analysis, the Bio-Plex Pro Cell Signaling Reagent Kit (BIO-RAD) was used. The treated lymphoma cells were cooled down by addition of the three-fold volume of ice-cold DPBS supplemented with 1xPhosSTOP (Roche) and 100μM sodium orthovanadate. After 5 minutes centrifugation (500xg, 4˚C) the cells were washed once. According to the instruction manual the cells were lysed in the provided buffer containing 1xfactor QG and 2mM PMSF. After shaking for 20 minutes at 4˚C, debris was removed by centrifugation (14000xg, 4˚C) for 15 minutes. The analysis with the Bio-Plex Protein Array system (BIO-RAD) was done as published before [26] and according to the manufacturer's instructions. Specific beads were used for p-SYK$^{Y352}$, p-ZAP70$^{Y319}$, p-Btk$^{Y223}$, p-AKT$^{S473}$, p-40S ribosomal protein S6$^{S235/S236}$, p-BAD$^{S136}$, p-MEK1$^{S217/S221}$, p-ERK1/2$^{T202/Y204, T185/Y187}$, p-p90RSK$^{S380}$, p-GSK3αß$^{S21/S9}$, p-HSP27$^{S78}$, p-JNK$^{T183/Y185}$, p-c-Jun$^{S63}$ and p-p65 NF-κB$^{S536}$. The Bio-Plex manager software and R package lxb was used for data acquisition.

## Network modeling

For the quantitative network modeling from systematic perturbation data we used our previously developed R package STASNet Version 1.0.2 (available on https://github.com/molsysbio/STASNet), which is a derivative of Modular Response Analysis [38, 63] adapted to model incomplete signaling perturbation data [26, 39]. Briefly, a perturbation $p$ in a biochemical network is propagated through the network interactions resulting in an experimentally observable global response $R_{ip}$ on node $i$ as steady state change. In contrast the local response

coefficient $r_{ip}$, represents the direct steady state change of node $i$ upon perturbation $p$ without allowing the perturbation to propagate through the network which is typically not measureable. Given a full systematic perturbation and measurements regime, these direct network connectivities can be estimated from the global response matrix as:

$$\boldsymbol{R} = -\boldsymbol{r}^{-1} * \boldsymbol{p} \tag{1}$$

Where $\boldsymbol{p}$ is a matrix containing the coefficients for direct target perturbation strengths, each column with only non-zero entries on the perturbed node(s). Due to noise and an incomplete experimental design, we can only numerically approximate the coefficients of the right side of Eq (1) from the global response matrix. As oftentimes not all nodes can be measured and perturbed, in STASNet we solve the underdetermined situation threefold: (i) we only fit entries of $\boldsymbol{R}$ that were measured, (ii) we utilize prior knowledge by using a literature-derived starting network (i.e., a binary representation of $\boldsymbol{r}$), with which we symbolically fill the entries of $\boldsymbol{r}$ and (iii) we apply gaussian elimination to derive identifiable combinations of $\boldsymbol{r}$. The parameters of the model are the entries of $\boldsymbol{r}$ and $\boldsymbol{p}$, termed local response coefficients, and perturbation coefficients, respectively. In here we use the term 'coefficients' for both, if identifiable, and if not the structurally identifiable combinations thereof will be termed 'coefficient paths'.

Regarding the perturbation coefficients we distinguish between stimulation and inhibition. We include the stimulated node $s$ in the network and attribute the stimulation strength to the local response coefficients directly reached by node $s$ and set the corresponding entry in $\boldsymbol{p}$ to 1. For modeling inhibitors, we cannot do this transfer as we model a twofold action on the downstream targets of the inhibited node: (i) for basal, i.e. unstimulated, signaling a negative effect on the activity of the downstream nodes ($l \in [-\infty, 0)$) and (ii) for simultaneous stimulation a dampening of the upstream signal by multiplying with the exponential of $l$ ($e^l \in (0,1)$), cf. [39] illustration in **Fig 2**.

We then quantify the so-called coefficient (path)s by a combination of Latin hypercube sample governed initialization and a Levenberg Marquardt gradient descent method to minimize the weighted quadratic difference between the measured global steady state changes and the model derived quantification (cf. Eq (2)). Afterwards we probe the network for superfluous or missing links and repeat the procedure for every step-wise development of the network. For a detailed mathematical explanation of the STASNet methodology we refer to supplementary information S1 in [39].

The model development strategy used in this study is illustrated in **Fig 1B** and accompanying text. More detailed modeling workflows as well as the required data for all modelled cell lines and scenarios are provided by step-by-step accompanying html reports available online (**S1 Text,** https://zenodo.org/doi/10.5281/zenodo.10732059).

To assess goodness of fit we calculate the weighted sum squared residuals (WSSR) which is generally defined as:

$$WSSR = \sum \left( \frac{data - model}{error} \right)^2 \tag{2}$$

where the data represents the data points, the model the prediction and the error the antibody-wise standard error derived from the replicate measurements.

Using Eq (2) for the model prediction of the fitted data (WSSR$_{fit}$) we calculated the reduced chi square statistics $X_r$ as goodness of fit assessment, which is defined as:

$$X_r = \frac{WSSR_{fit}}{n_{free\ parameter}}$$ (3)

Where the number of free parameters is given by number of data points minus the number of model parameters, i.e., identifiable coefficient (path)s. For the model consistency step the STASNet function *simulateModel()* was used to generate the prediction of unstimulated but inhibited data of which the sum of weighted squared residuals (WSSR$_{simulation}$) was calculated. We then compared this to the residuals that a suitable null model, in here unperturbed data, would produce (WSSR$_{nullmodel}$) and defined the prediction capacity better than null model as follows:

$$Error\ reduction(\%) = 100 * \left( \frac{WSSR_{nullmodel} - WSSR_{simulation}}{WSSR_{nullmodel}} \right)$$ (4)

For comparative combined modeling we used the following strategy. Since inhibitor dosages were always the same but the response to the inhibitors varied from cell line to cell line, the actual difference is thought to happen on the internal wiring not on the inhibitor perturbation strength coefficient. Therefore, when comparing different models, we fixed the coefficients encoding the inhibitor strengths to the same value across the models to be compared and allowed the STASNet algorithm to compensate by refitting the remaining coefficient (path)s using the STASNet routine *refitModel()*.

As the typical kinase inhibitors usually block the phosphorylation of downstream nodes, by either interfering with the binding of targets or ATP, the inhibitor action is implemented such that it affects the downstream nodes of the actual target but not the target itself. In case this effect is also acting on the targeted node itself we will ignore the measurement of the targeted node (which is done for inhibitors of AKT and Btk in this study, see also **S2** and **S4** **Figs** and **S1 Text**).

## Mass spectrometry based phosphoproteomics

For phosphoproteomic profiling by mass spectrometry, cell pellets were lysed in urea lysis buffer (6M urea, 2M thiourea, 100mM Tris-HCl, pH 8, 150mM NaCl, 1mM EDTA, phosphatase inhibitor cocktail 2 and 3, 10mM NaF, Sigma), reduced with 10mM DTT (dithiothreitol, Sigma) for 45 minutes followed by alkylation with 40mM CAA (2-chloroacetamide, Sigma) for 30 minutes. After treatment with Benzonase (Merck, 50 units) for 30 min at 37˚C, samples were centrifuged for 10 minutes at 12 000 rpm. The supernatant was collected and protein concentration was determined. 200µg protein per sample was digested with 2µg endopeptidase LysC (Wako), followed by a 3:1 dilution with 100mM ammonium bicarbonate and addition of 2µg sequence-grade trypsin (Promega). Samples were digested at room temperature overnight and acidified with formic acid (final concentration 1%). The resulting peptides were cleaned using C18 SepPak columns (Waters, 100mg/1cc), dried and resolved in 50mM HEPES (pH 8). Peptides were labeled with 11-plex tandem mass tag (TMT, Fisher Scientific) reagents following the vendors instructions. After combining all samples and C18 SepPak-based clean-up (Waters, 200mg/1cc), samples were fractionated by high-pH reversed phase off-line chromatography (1290 Infinity, Agilent) and pooled into 15 fractions, which were applied to IMAC based phosphopeptide enrichment as described [53]. For LC-MS/MS measurements, peptides were reconstituted in 3% acetonitrile with 0.1% formic acid and separated on a reversed-phase column (20 cm fritless silica microcolumns with an inner diameter of 75 µm, packed with

ReproSil-Pur C18-AQ 1.9 μm resin (Dr. Maisch GmbH)) using a 98 min gradient with a 250 nl/min flow rate of increasing Buffer B (90% ACN, 0.1% FA) concentration (from 2% to 60%) on a High Performance Liquid Chromatography (HPLC) system (Thermo Fisher Scientific) and analyzed on a Q Exactive Plus instrument (Thermo Fisher Scientific). The mass spectrometer was operated in data-dependent acquisition mode using the following settings: full-scan automatic gain control (AGC) target 3 x $10^6$ at 70K resolution; scan range 350–2000 m/z; Orbitrap full-scan maximum injection time 10ms; MS/MS scan AGC target of 5 x $10^4$ at 35K resolution; maximum injection time 100ms; normalized collision energy of 32 and dynamic exclusion time of 30s; precursor charge state 2–6, ten MS2 scans per full scan. RAW data were analyzed with MaxQuant software package (v 1.6.0.1) using the Uniprot databases for human (2018–05). The search included variable modifications of methionine oxidation, N-terminal acetylation, deamidation (N and Q) and phosphorylation (STY) and fixed modification of carbamidomethylated cysteine. Reporter ion MS2 for TMT11 was selected (internal and N-terminal) and TMT batch specific corrections factors were specified. The FDR (false discovery rate) was set to 1% for peptide and protein identifications. Unique and razor peptides were included for quantification.

## Statistical analysis and evaluation of Tandem Mass Tag data

After excluding reverse database hits and potential contaminants the resulting list of phosphosites was filtered for localisation probability (>0.75) and unified for ambiguously mapped phosphosites leaving 28871 phosphosites for downstream analysis. After quantile normalisation, differentially regulated sites were identified using the limma R package (v. 3.54.0) and resulting p-values were corrected for multiple testing using Benjamini-Hochberg method (FDR< = 0.05).

Kinase enrichment of phosphopeptides was conducted by using information from PhosphoSitePlus (https://www.phosphosite.org, Kinase_Substrate_Dataset 24.11.2021) to identify putative kinases that regulate the phosphosites. Then for all kinases that regulate at least 5 phosphosites in our data set, we tested with a two-sided paired t-test whether the mean fold change to the reference treatment (i.e., Control+DMSO for α-IgM+DMSO and α-IgM +DMSO for α-IgM+inhibitor) was significantly different from 0. Afterwards we corrected for multiple-testing using the Benjamini-Hochberg approach.

To assess the effect of treatments on the total proteome we in parallel analyzed the total protein data (n = 8785) analogously to the phosphoproteome data but without localization probability filter. In total only 71 proteins were found significantly changed for any treatment of which 35 overlapped with significant phosphosite changes, affecting 1.1% of proteins whose phosphoproteome significantly changed and 1.6% of significant phospho-signals (n = 141). Due to the little change of total proteins we decided to only concentrate on the phosphoproteome in this study.

## Supporting information

**S1 Fig. Changes of selected phosphosites over time after treatment of BL-2 cells with α-IgM indicate a (quasi) steady state at 30 min.** The phosphorylation changes were estimated by (A) immunoblot and (B) Bead-based ELISA analysis on the same samples depicted in fluorescence intensities.
(TIF)

**S2 Fig. BL-2 data and model fits on network structures learned from indicated cell lines.** Heatmaps of mean log2 fold changes (n = 3) to untreated for model input data for BL-2 (BL-2

data) and model results for best network structures found for BL-2 and BL-41 data when adjusting the response coefficients but not the network structure to BL-2 data. Blanks in BL-2 data are withhold data which act contrary to central model assumptions that inhibitors reduce phosphorylations of downstream targets not phosphorylation of their targets and are therefore not modelled (see Material and Methods). Blanks in BL-2 model are due to the removal of the edge connecting to HSP27 in the final model structure. For more information see **S1 Text** BL-2_network_model.html: Tab '11. Rem. PI3K -> Btk' and BL-41_network_model.html: Tab 'Model transfer to BL-2'.
(TIF)

**S3 Fig. Network comparison of individually derived models on BL-2 and BL-41 cell data.** Literature-derived starting network (cf. **Fig 1C**) and adjustments during model development for individual models for the indicated cell lines. Numbers indicate added (green) and removed (grey) links.
(TIF)

**S4 Fig. BL-41 data and model fits on network structures learned from indicated cell lines.** Heatmaps of mean log2 fold changes (n = 3) to untreated for model input data for BL-41 (BL-41 data) and model results for best network structures found for BL-2 and BL-41 data when adjusting the coefficient (path)s but not the network structure to BL-41 data. Blanks in BL-41 data are withhold data which act contrary to central model assumptions that inhibitors reduce phosphorylations of downstream targets not phosphorylation of their targets and are therefore not modelled (see Materials and Methods). Blanks in BL-2 model are due to the removal of the edge connecting to HSP27 in the final model structure. For more information see **S1 Text** BL-41_network_model.html: Tab '9. Rem. Syk -> PI3K' and BL-2_network_model.html: Tab 'Model transfer to BL-41' variant *BL2-model as initial model.*
(TIF)

**S5 Fig. p38α/MAPK14 is one of the most abundantly expressed isoforms of p38 in BL-2.** Total reads of the p38 subunits α (MAPK14), β (MAPK11), γ (MAPK12), δ (MAPK13) are displayed from RNA sequencing analysis of BL-2 cells (n = 3, data from microarrays conducted at [64]).
(TIF)

**S6 Fig. Increased MEK/ERK-pathway activity after p38 MAPK14 intervention can be observed also in response to CD40-Ligand treatment in BL cells.** CA-46 BL cells were treated with CD40 ligand for up to 90 min without or with 2μM p38 inhibitor SB203580. Data were analyzed by classical IB-chemoluminescence imaging and by the Fusion-FX platform.
(TIF)

**S7 Fig. Relative growth of DLBCL cells after drug treatments.** Cell counts of indicated DLBCL cell lines treated for 72h with indicated inhibitors and concentrations, as log2 fold change to respective mean solvent control (DMSO). P-values derived by two-sided T-test; n = 3. Inhibitors (target): SB203580 (p38), AZD6244 (MEK), Rapamycin (mTORC1), BKM120 (PI3K) and MLN120B (IKK).
(TIF)

**S1 Table. Side-by-side coefficient (path)s and confidence intervals for BL-2 and BL-41 data with fixed inhibitor quantifications.** Parametrization for the best fit and in brackets the upper and lower boundaries of the 95% confidence interval, derived by profile likelihood (alpha = 0.05, 1 degree of freedom). **ni** denotes nonidentifiable confidence intervals in that direction (i.e., alteration of the coefficient can be compensated by changing other model

coefficients). Coefficient (path)s with non-overlapping confidence intervals are termed significantly different between BL-2 and BL-41. Coefficient term definition: r_*source_target*.
(DOCX)

**S2 Table. Side-by side model coefficient path(s) and confidence intervals for HBL-1 and OCI-LY3 data with fixed inhibitor quantifications.** Parametrization for the best fit and in brackets the upper and lower boundaries of the 95% confidence interval, derived by profile likelihood (alpha = 0.05, 1 degree of freedom). **ni** denotes nonidentifiable confidence intervals in that direction (i.e., alteration of the coefficient can be compensated by changing other model coefficients). Coefficient (path)s with non-overlapping confidence intervals are termed significantly different between HBL-1 and OCI-LY3. – indicates missing link in either cell line. Coefficient term definition: r_*source_target*.
(DOCX)

**S1 Text. Overview of Online Supplements.** (https://zenodo.org/doi/10.5281/zenodo.10732059): BL-2_network_model.html. BL-41_network_model.html. DLBCLs_network_model.html. global_extend.R.
(DOCX)

## Acknowledgments

We thank all members of the joint project MMML-Demonstrator for helpful discussion in running the analysis.

## Author Contributions

**Conceptualization:** Bertram Klinger, Isabel Rausch, Arnd Kieser, Nils Blüthgen, Dieter Kube.

**Data curation:** Bertram Klinger, Isabel Rausch, Anja Sieber, Marieluise Kirchner, Dieter Kube.

**Formal analysis:** Bertram Klinger.

**Funding acquisition:** Bertram Klinger, Arnd Kieser, Nils Blüthgen, Dieter Kube.

**Investigation:** Isabel Rausch, Anja Sieber, Helmut Kutz, Vanessa Kruse, Marieluise Kirchner, Arnd Kieser, Dieter Kube.

**Methodology:** Bertram Klinger.

**Project administration:** Nils Blüthgen, Dieter Kube.

**Resources:** Philipp Mertins, Arnd Kieser, Nils Blüthgen, Dieter Kube.

**Software:** Bertram Klinger.

**Supervision:** Philipp Mertins, Arnd Kieser, Nils Blüthgen, Dieter Kube.

**Visualization:** Bertram Klinger, Isabel Rausch, Dieter Kube.

**Writing – original draft:** Bertram Klinger, Arnd Kieser, Nils Blüthgen, Dieter Kube.

**Writing – review & editing:** Bertram Klinger, Nils Blüthgen, Dieter Kube.

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
