## [Decision Letter · Decision Letter 0]

10 Jun 2024

Dear Mr. Klinger,

Thank you very much for submitting your manuscript "Quantitative modelling of signaling in aggressive B cell lymphoma unveils conserved core network" for consideration at PLOS Computational Biology.

As with all papers reviewed by the journal, your manuscript was reviewed by members of the editorial board and by several independent reviewers. In light of the reviews (below this email), we would like to invite the resubmission of a significantly-revised version that takes into account the reviewers' comments.

We cannot make any decision about publication until we have seen the revised manuscript and your response to the reviewers' comments. Your revised manuscript is also likely to be sent to reviewers for further evaluation.

Sincerely,

Marc R Birtwistle, PhD

Section Editor

PLOS Computational Biology

Stacey Finley

Section Editor

PLOS Computational Biology

Reviewer's Responses to Questions

**Comments to the Authors:**

Reviewer #1: The authors present new models of B cell lymphoma using their Boolean Nested Effects Modelling (B-NEM) approach. Compared to previous models of BCR signaling, the presented models have more ample scope, i.e. covers multiple pathways and their cross talks, and is (semi-)quantitative, therefore allowing to better study also complex network mechanisms such as feedback and cross-talks. The model is derived starting from literature-derived networks and is trained using in house targeted phosphoproteomics (magnetic bead-based

multiplex immunoassay) data to add and remove links and estimate parameter values using their R package STASNet. The most impactful result of the paper is that, using computational modeling on two Burkitt lymphoma cell lines, the authors identify important regulatory mechanisms not yet described in BCR activated signaling. These pathways crosstalks were extensively validated and characterized experimentally using westerns blot and global phosphoproteomics. Additionally, the authors identify a core network that is shared also with Diffuse large B cell lymphoma cell lines, while also highlighting some differences. Overall, these results can help us to better understand B cell lymphoma aberrations with possible future clinical implications. This is a very interesting paper with a very rigorous modeling development and impactful biological insights.

Here below some comments:

1. I might not be reading the data right, but why does the MEK inhibitor in the presence of IgM stimulation induce MEK activation in all 4 cell lines (e.g. for BL-2 in stimulated condition MEK has activity = 2.6 in the uninhibited condition and 4.6 in the MEKi condition)? For BL-41 this is the case also for the unstimulated condition (MEK activity in MEKi = 0.3)

2. Would be interesting to investigate what are the main differences between the BL-2 and BL-45 data (e.g. with a simple scatterplot, and possibly also with the DLBLC cell lines). The BL-41 data seems easier to predict: the reduced chi-square for the BL-2 even after BL-2 specific training (adjusted) is worse than the one obtained for BL-41 with the original literature network. Do the author know what makes the BL-2 data more difficult to predict? It would be interesting to visualize also the model fit to each experimental condition (as supplementary).

3. I understand why the authors use the unstimulated data as 'test' set when looking separately at performances of the BL-2 and BL-41 models to ensure consistency. However, if the aim is to have the best model that can generally describe BL, those data could provide very useful for model training. Since the authors are anyway testing generalizability of the trained models on different cell lines, I would find it interesting to train the models also with the full data (stimulated and unstimulated) for each cell line.

4. Do I understand correctly that no replicates were performed for the Bio-Plex multiplex immunoassay? Can the authors be confident that the differences between cell lines is not driven by measurement error?

5. Why is the added negative feedback from p38 to RAF1 instead of directly to MEK? I would imagine that for the model if it difficult to distinguish between the two since RAF1 is not perturbed nor measured in the data used for model training. Wouldn't the data in Figure 3 suggest that the feedback mechanism is not mediated by RAF1?

Minor:

6. Since MRA is not broadly known, it would be useful to have a short section describing the principles of the MRA, e.g. what is the intuition behind the model formalisms, what are the model parameters.

7. How are inhibitors effects simulated? In page 8 the authors specify that "fixed the parameters encoding the inhibitor strengths" but never mention how simulations of these models intuitively work and/or what are the model parameters.

8. Check order of citation of Figures, e.g. figure 1A never cited in text, panel D cited before panel C

9. On page 20 the authors cite FIGURE 3F instead of FIGURE 3E

Reviewer #2: The manuscript describes modeling of BCR signaling using systematically perturbed phosphorylation data in two types of cells: BL-2 and BL-41 cells. The authors use the software package STASNet (STeady-STate Analysis of Signalling Networks) that was described by the team of Nils Blüthgen in 2018. This package is based on Modular Response Analysis (MRA) method that was developed by Kholodenko’s team to infer the response network from the steady-state global responses to perturbations. The authors start from literature-based models and use perturbation data for one cell line to create a model for two cell lines and reveal network features of the BCR signaling network, validated experimentally using phospho-proteomics assays. The manuscript is well-written and provides a good biological background and motivation for modeling and analysis. Two major issues are:

- Not enough background on the modeling techniques used, and some explanations are not clear.

- Supplemental web pages provide a lot of additional information clarifying the modeling and analysis process, but these pages are not described in detail and not referred to from the manuscript in a standard way (e.g. Figure S10, etc).

Specifically:

1. The word semi-quantitative modeling is never defined. Even the notion of steady state modeling is never mentioned. I had to go all the way down to Kholodenko et al., 2002 to understand what is semi-quantitative. The authors need at least one paragraph describing the terms semi-quantitative modeling (it’s not commonly used), MRA, and STASNet. Figure 1A-B is a nice introduction but it is not enough, a figure like Fig .2 in Dorel et al., 2018 would be helpful to understand the underlying mechanism and what are parameters in the model.

2. The authors never describe what are parameters to be fitted, and they use the terms “model parameters”, “strength parameters”, “network coefficients” and “pathway coefficients” without any explanation. Is it the same as “response coefficient” in STASNet? It needs to be defined and used consistently.

3. The authors use the word “stronger”. Is it equivalent to “larger in absolute value”? Specifically, they say “the parameter corresponding to the pathway from Btk via p38 to MEK is estimated to be stronger in BL-41 than in BL-2 cells”. Is stronger in this sentence means larger in absolute value (it may be negative)?

4. “Unseen nodes” need to be better described. It’s really confusing: the authors have some nodes being unmeasured, and some nodes being measured but not used for network construction.

5. Model/network adaptation with sequential adjustment steps are apparently described in the supplemental web pages. It would be much clearer if the authors directly refer to the webpage “BL-2 network model development” when talking about 7 and 5 adjustments, etc. For example, the paragraph on page 10 states: “During model adaption, 4 adjustments were shared between BL-2- and BL-41. This includes, two of the three additional links already identified in BL-2 cells before, i.e., p38->RAF and mTORC1->JNK. However, 7 and 5 adjustments were BL-2- and BL-41- specific, respectively (FIGURE 2C). This divergence in proposed network structure also manifests in the fact that only 14 of the 21 identifiable path coefficients were shared in both cell lines.” It’s absolutely not clear what are these 7 adjustments and what are these 5 adjustments. The figure 2C is just a Venn diagram with numbers 0,1,2,5 and 6. What are these 14 path coefficients is not clear. The webpage describes 31 significant link additions when the network is transferred to BL-41 cell line, but these 31 links do not correspond to anything in the manuscript.

In conclusion, it’s a very interesting methodology, model, and results, but a better description of the modeling and analysis process is required. Perhaps supplemental material guiding through supplemental web pages could do the job.

**Have the authors made all data and (if applicable) computational code underlying the findings in their manuscript fully available?**

Reviewer #1: Yes

Reviewer #2: Yes

PLOS authors have the option to publish the peer review history of their article (what does this mean?). If published, this will include your full peer review and any attached files.

Reviewer #1: No

Reviewer #2: No
---

## [Decision Letter · Decision Letter 1]

12 Sep 2024

Dear Mr. Klinger,

We are pleased to inform you that your manuscript 'Quantitative modeling of signaling in aggressive B cell lymphoma unveils conserved core network' has been provisionally accepted for publication in PLOS Computational Biology.

Best regards,

Marc R Birtwistle, PhD

Section Editor

PLOS Computational Biology

Marc Birtwistle

Section Editor

PLOS Computational Biology

Reviewer's Responses to Questions

**Comments to the Authors:**

Reviewer #1: The authors addressed all my concerns. The paper is now more clear including also a more extensive description of their modelling approach. I also appreciate the new analysis comparing the models for the different cell lines that makes the story more complete. This paper is a very nice example of how mathematical modeling of signaling pathways can help to elucidate underlying mechanisms and deregulations in cancer.

Reviewer #2: The authors successfully responded to all comments.

**Have the authors made all data and (if applicable) computational code underlying the findings in their manuscript fully available?**

Reviewer #1: Yes

Reviewer #2: Yes

PLOS authors have the option to publish the peer review history of their article (what does this mean?). If published, this will include your full peer review and any attached files.

Reviewer #1: **Yes: **Federica Eduati

Reviewer #2: No

---

## [Editor Report · Acceptance letter]

24 Sep 2024

PCOMPBIOL-D-24-00505R1 

Quantitative modeling of signaling in aggressive B cell lymphoma unveils conserved core network

Dear Dr Klinger,

I am pleased to inform you that your manuscript has been formally accepted for publication in PLOS Computational Biology. Your manuscript is now with our production department and you will be notified of the publication date in due course.

With kind regards,

Anita Estes
